# 4-Levels Vertically Stacked SiGe Channel Nanowires Gate-All-Around Transistor with Novel Channel Releasing and Source and Drain Silicide Process

**DOI:** 10.3390/nano12050889

**Published:** 2022-03-07

**Authors:** Xiaohong Cheng, Yongliang Li, Fei Zhao, Anlan Chen, Haoyan Liu, Chun Li, Qingzhu Zhang, Huaxiang Yin, Jun Luo, Wenwu Wang

**Affiliations:** Integrated Circuit Advanced Process Center, Institute of Microelectronics, University of Chinese Academy of Sciences, Beijing 100029, China; chengxiaohong@ime.ac.cn (X.C.); zhaofei@ime.ac.cn (F.Z.); chenanlan@ime.ac.cn (A.C.); liuhaoyan@ime.ac.cn (H.L.); lichun@ime.ac.cn (C.L.); zhangqingzhu@ime.ac.cn (Q.Z.); yinhuaxiang@ime.ac.cn (H.Y.); luojun@ime.ac.cn (J.L.); wangwenwu@ime.ac.cn (W.W.)

**Keywords:** SiGe channel, gate-all-around, release, silicide

## Abstract

In this paper, the fabrication and electrical performance optimization of a four-levels vertically stacked Si_0.7_Ge_0.3_ channel nanowires gate-all-around transistor are explored in detail. First, a high crystalline quality and uniform stacked Si_0.7_Ge_0.3_/Si film is achieved by optimizing the epitaxial growth process and a vertical profile of stacked Si_0.7_Ge_0.3_/Si fin is attained by further optimizing the etching process under the HBr/He/O_2_ plasma. Moreover, a novel ACT@SG-201 solution without any dilution at the temperature of 40 °C is chosen as the optimal etching solution for the release process of Si_0.7_Ge_0.3_ channel. As a result, the selectivity of Si to Si_0.7_Ge_0.3_ can reach 32.84 with a signature of “rectangular” Si_0.7_Ge_0.3_ extremities after channel release. Based on these newly developed processes, a 4-levels vertically stacked Si_0.7_Ge_0.3_ nanowires gate-all-around device is prepared successfully. An excellent subthreshold slope of 77 mV/dec, drain induced barrier-lowering of 19 mV/V, I_on_/I_off_ ratio of 9 × 10^5^ and maximum of transconductance of ~83.35 μS/μm are demonstrated. However, its driven current is only ~38.6 μA/μm under V_DS_ = V_GS_ = −0.8 V due to its large resistance of source and drain (9.2 × 10^5^ Ω). Therefore, a source and drain silicide process is implemented and its driven current can increase to 258.6 μA/μm (about 6.7 times) due to the decrease of resistance of source and drain to 6.4 × 10^4^ Ω. Meanwhile, it is found that a slight increase of leakage after the silicide process online results in a slight deterioration of the subthreshold slope and I_on_/I_off_ ratio. Its leakage performance needs to be further improved through the co-optimization of source and drain implantation and silicide process in the future.

## 1. Introduction

The vertically stacked horizontal gate-all-around (GAA) transistors are now established as the most promising candidate to the FinFETs in sub-5nm technology node, due to the excellent electrostatic and short channel control [1,2,3]. Moreover, to keep Moore’s Law alive as long as possible, researchers are also looking for alternatives to silicon channel material, like SiC, GaN, SiGe and Ge [4,5,6,7]. Among them, SiGe materials, especially those with Ge concentration between 20% and 40%, have been considered as the channel material of GAA devices. This is because they have higher electron and hole mobility, better negative bias temperature instability (NBTI) reliability [8,9] than Si and are more compatible with present Si platform [9,10,11]. However, the fabrication of stacked SiGe nanowire/nanosheet (NW/NS) GAA devices still face many challenges, such as a high-quality stacked SiGe/Si fin structure preparation, high selectively SiGe NW/NS release, inner spacer, source/drain (S/D) epitaxial process, etc. [11,12,13,14,15]. These processes are critical for the preparation of the stacked SiGe channel NW/NS device. Prior to this work, there are several reports on the preparation of SiGe/Si fin in terms of the epitaxial growth and dry etching process of stacked SiGe/Si [11,12,16,17]. In addition, the releasing technologies of SiGe nanowire by highly selective removal of Si have been investigated in the last years using the dry etching with CF4-based plasma [18], in situ HCl gaseous thermal etching [19] and wet chemical etching [13,20]. However, the former two methods are normally conducted using specific tools to avoid damages or selectivity issues. Meanwhile, wet chemical etching is an easier strategy since it can achieve a high selective etching only by choosing an appropriate alkaline solution and optimizing its temperature. Moreover, the wet chemical etching can be implemented in a common wet bench or spin on single wafer tool. Recently, a new alkaline solution, ACT@SG-201, was proposed to remove Si sacrificial layer and inhibit the loss of SiGe layer [21]. Therefore, for a 4-levels vertically stacked Si_0.7_Ge_0.3_ NW GAA device, a high crystalline quality epitaxial growth and vertical dry etching of stacked Si_0.7_Ge_0.3_/Si multilayer and a high selective release of stacked SiGe channels need to be further studied, as only a few studies have been reported so far.

In this paper, a 4-levels vertically stacked Si_0.7_Ge_0.3_ NWs GAA p-MOS device is demonstrated by developing the fabrication process of a high-quality stacked Si_0.7_Ge_0.3_/Si fin and developing the releasing process of the stacked Si_0.7_Ge_0.3_ channels. Furthermore, its driven current can be further enhanced by introducing a process of nickel-based silicide at S/D regions.

## 2. Materials and Methods

The process flow that was used for the fabrication of the stacked Si_0.7_Ge_0.3_ NWs GAA p-MOS transistor is illustrated in Figure 1. After the n-Well formation (Figure 1a), a vertical stacked Si_0.7_Ge_0.3_/Si fin was fabricated by developing a high-quality epitaxial growth and anisotropic dry etching process used the HBr-based plasma under the sidewall image transfer technique (Figure 1b). After that, the shallow trench isolation (STI), dummy gate and spacers were defined, and *p*-type BF_2_ ions were implanted in Si_0.7_Ge_0.3_/Si fins to form S/D regions (Figure 1c). In addition, some samples employed a self-aligned S/D nickel-based silicide process [22,23] (Figure 1d). The stacked Si_0.7_Ge_0.3_ NWs were subsequently released in the replacement metal gate (RMG) module by using an optimized ACT@SG-201 solution for the selective removal of Si sacrificial layers. Additionally, the Al_2_O_3_/HfO_2_ bi-layer high-k (HK) dielectric and TiN-based/W metal gate (MG) stack were used as the HK/MG stack (Figure 1e). Finally, the four terminals were formed by the normal contact and metal connections process (Figure 1f).

The film quality and device structure were observed using high-resolution transmission electron microscopy (HRTEM) and the high angle annular dark field scanning transmission electron microscope (HAADF-STEM). The fin etch and selective etch profiles were performed by scanning electron microscopes (SEM). The Energy-dispersive X-ray spectroscopy (EDX) mapping analysis was employed to verify elements of the final stacked Si_0.7_Ge_0.3_ NWs GAA device. The electrical characterization was performed using a Keithley 4200 semiconductor parameter analyzer.

## 3. Results and Discussion

### 3.1. High-Quality Stacked Si_0.7_Ge_0.3_/Si Fin Formation

A vertical stacked Si_0.7_Ge_0.3_/Si fin with a high crystalline quality is one of key factors for the fabrication of a stacked Si_0.7_Ge_0.3_/Si NWs GAA device. As shown in Figure 2a,b, a high-quality epitaxial growth of stacked Si_0.7_Ge_0.3_/Si multilayer was verified using the HRTEM and HAADF-STEM analysis. No threading dislocation defects, as well as distinct and sharp interfaces between the Si_0.7_Ge_0.3_ and Si, can be found. Meanwhile, the thickness of Si_0.7_Ge_0.3_ from top to bottom is 8.3, 8.2, 8.1 and 10.1 nm under the same time of epitaxial growth. In other words, the thickness of bottom Si_0.7_Ge_0.3_ is ~2 nm thicker than that of others. It is known that the epitaxial rate is strongly dependent on the crystallization of the under-layer, that is, the epitaxial rate might be decreased if multi-crystallization occurs in the under-layer [9]. At the same time, the thickness of Si is measured as 13.5, 12.7, 12.0 and 12.3 nm from top to bottom for the better release of Si_0.7_Ge_0.3_ NW channel. Moreover, it can be seen that the thickness of top Si is ~1 nm thicker than that of others. Its purpose is to increase the process window for the following poly and spacer etching process.

To get a uniform stacked Si_0.7_Ge_0.3_ NW channel length and excellent gate control, a vertical profile of stacked Si_0.7_Ge_0.3_/Si fin is a critical process. Based on previous etching results of the SiGe fin and two period stacked SiGe/Si fin [11,17], the HBr/O_2_/He plasma was applied for four-period-stacked Si_0.7_Ge_0.3_/Si fin, as shown in Figure 2c. It is found that a not very vertical profile of stacked Si_0.7_Ge_0.3_/Si fin was attained. To further optimize its etching profile, the bias voltage of fin etching was fine-tuned from −90 V to −100 V. A vertical four-period stacked Si_0.7_Ge_0.3_/Si fin structure was attained by increasing ions bombarding, as shown in Figure 2d.

### 3.2. Si_0.7_Ge_0.3_ NW Channel Release

For a selective removal of Si to Si_0.7_Ge_0.3_, the wet etching approach using an alkaline solution is still a better choice. For example, the conventional TMAH solution, after the co-optimization of concentration and temperature, can be chosen to selectively remove the Si to Si_0.7_Ge_0.3_ with a selectivity of ~17.3 [13]. To further enhance the selectivity of the wet etching process, a novel ACT@SG-201 solution is employed to verify the release process of four-levels vertically stacked Si_0.7_Ge_0.3_ NW channel. This novel solution has effective Si surface modifier and SiGe corrosion inhibitor to achieve a relatively high selectivity [21].

Firstly, the effect of temperature on the etching characteristics was verified for the ACT@SG-201. Figure 3a,b present the etching rates of Si, Si_0.7_Ge_0.3_ and selectivity of Si to Si_0.7_Ge_0.3_ under ACT@SG-201 solution at temperatures of 20 °C, 40 °C and 60 °C. The lateral etching rate of Si is calculated based on the tunnel depth divided by immersion time. Additionally, the vertical etching rate of Si_0.7_Ge_0.3_ is calculated based on the thickness loss of Si_0.7_Ge_0.3_ per side at the edge position divided by immersion time [13]. It can be found that the vertical etching rate of Si_0.7_Ge_0.3_ layers increase with the increase of temperature, but the lateral etching rate of Si layers increases first and then decrease with the increase of temperature. Therefore, the selectivity of Si to Si_0.7_Ge_0.3_ increases from 28.67 to 32.84 as the temperature of ACT@SG-201 solution increase from 20 to 40 °C and then decrease to 27.76 as the temperatures of ACT@SG-201 solution further increase to 60 °C. It can be concluded that ACT@SG-201 at 40 °C is the optimal temperature for the selective removal of Si to Si_0.7_Ge_0.3_.

At the same time, the etching rate of Si, Si_0.7_Ge_0.3_ and selectivity of Si to Si_0.7_Ge_0.3_ with different concentration of ACT@SG-201 solution at 40 °C are presented in Figure 4a,b. As we can see that the etching rate of Si and Si_0.7_Ge_0.3_ are decreasing with the increasing of the concentration of ACT@SG-201. However, the selectivity of Si to Si_0.7_Ge_0.3_ increase from 20.67 to 32.84 because the etching rate of Si_0.7_Ge_0.3_ decreases more than that of Si. Based on the above results, the ACT@SG-201 without any dilution at the temperature of 40 °C is chosen as the optimal etching solution for the selective etching of Si to Si_0.7_Ge_0.3_.

Moreover, the selective etching profile of two periods Si_0.7_Ge_0.3_/Si multilayer by using the optimal ACT@SG-201 at 40 °C is presented in Figure 5. The Si_0.7_Ge_0.3_ layers extremities are “rectangular” without significant Si_0.7_Ge_0.3_ loss after etching for 10 min at 40 °C due to its high selectivity. This result further confirms that ACT@SG-201 without any dilution at the temperature of 40 °C is the optimal condition due to its high selectivity.

### 3.3. 4-Levels Vertically Stacked Si_0.7_Ge_0.3_ NW GAA Device

Based on above newly developed fabrication process of high-quality stacked Si_0.7_Ge_0.3_/Si fin, and release process of Si_0.7_Ge_0.3_ NW channel, a four-levels vertically stacked Si_0.7_Ge_0.3_ NWs GAA device is prepared successfully. Figure 6 shows the cross-sectional image of the Si_0.7_Ge_0.3_ NW channels area under the HK/MG stack at the end of fabrication processing. It can be found that vertical and uniform four-levels stacked Si_0.7_Ge_0.3_ channels with almost the same width (~13.6 nm) were achieved. This result indicates that our newly developed fin formation and NWs release process are effective for the fabrication of vertically stacked Si_0.7_Ge_0.3_ NWs GAA device. In addition, the EDX mapping results of the four-levels vertically stacked Si_0.7_Ge_0.3_ NWs GAA device under the HK/MG stack at the end of the fabrication processing are shown in Figure 6b–f. From Figure 6b,c, it can be seen that uniform Ge and Si elements are distributed in the channel area. Namely, the stacked channels are Si_0.7_Ge_0.3_ layers and the Si sacrificial layers have been completely removed. These results further confirmed that our newly developed fin formation and NWs release process are effective for the fabrication of vertically stacked Si_0.7_Ge_0.3_ NW GAA device. Meanwhile, the stacked Si_0.7_Ge_0.3_ NW channels were well surrounded by the ALD HK/MG stacks to form a GAA structure, as shown in Figure 6d–f, which could provide an excellent gate control ability for the four-levels vertically stacked Si_0.7_Ge_0.3_ NW GAA device.

The typical I_DS_-V_GS_, g_m_-V_GS_ and I_DS_-V_DS_ characteristics of the four-levels vertically stacked Si_0.7_Ge_0.3_ NWs GAA device are shown in Figure 7a–c, respectively. An excellent subthreshold slope (SS) of ~77 mV/dec, low drain induced barrier-lowering (DIBL) of ~19 mV/V, high I_on_/I_off_ ratio of ~9 × 10^5^ and the maximum of transconductance (g_m,max_) of ~83.35 μS/μm were demonstrated using the above newly developed process. In particular, its excellent SS performance indicates that a good electrostatic control is obtained for the 4-levels vertically stacked Si_0.7_Ge_0.3_ NWs GAA device. However, its driven current (I_on_) was only ~38.6 μA/μm under V_DS_ = V_GS_ = −0.8 V due to its large resistance of source and drain (R_SD_). According to the I_DS_-V_DS_ curves of different device widths at liner region, it can be calculated that the value of R_SD_ can reach 9.2 × 10^5^ Ω. This is because the direct connect was formed between contact and Si_0.7_Ge_0.3_/Si fin at S/D area without epitaxial or silicide process.

### 3.4. Electrial Perforemance Optimization

To further improve the drive current and reduce the R_SD_, a S/D silicide process is proposed to fabricate the four-levels vertically stacked Si_0.7_Ge_0.3_ NWs GAA device. The HAADF-STEM and EDX mapping analysis were employed to check the S/D region of the four-levels vertically stacked Si_0.7_Ge_0.3_ NWs GAA device. The results are shown in Figure 8a–d. It is found that the top first period and partial second period Si_0.7_Ge_0.3_/Si fin at S/D area have been silicified with a uniform and smooth interface. Meanwhile, the EDX mapping results, as shown in Figure 8b–d, proved that the uniform Ni-Si or Ni-SiGe silicide was formed on the top of the Si_0.7_Ge_0.3_/Si fin at S/D area.

The I_DS_-V_GS_, g_m_-V_GS_ and I_DS_-V_DS_ characteristics of 4-levels vertically stacked Si_0.7_Ge_0.3_ NWs GAA device with S/D silicide process are shown in Figure 9a–c, respectively. Compared with device without S/D silicide process, the I_on_ can increase from 38.6 to ~258.6 μA/μm (about 6.7 times), as well as the g_m,max_ can increase from 83.35 to 562.73 μS/μm (about 6.7 times), because its R_SD_ can reduce from 9.2 × 10^5^ to 6.4 × 10^4^ Ω by employing the S/D silicide process. These results verified that the S/D silicide process can improve the I_on_ of the 4-levels vertically stacked Si_0.7_Ge_0.3_ NW GAA device by reducing its R_SD_.

To check if this process has other side effects, the key electrical parameters of devices with or without S/D silicide are compared in detail. The results are summarized in Table 1. Although its I_on_, g_m,max_ and R_SD_ had obviously improvement, its other parameters, such as SS, DIBL, leakage (I_off_) and I_on_/I_off_ ratio were slightly worse. For example, the leakage increased from 4.22 × 10^−5^ to 3.71 × 10^−4^ μA/μm, SS increased from 77 to 93 mV/dec, and its I_on_/I_off_ ratio decreased from 9 × 10^5^ to 7 × 10^5^. We believe that the slight deterioration of SS and I_on_/I_off_ ratio is caused by the increase of I_off_. Its leakage performance needs to be further improved through the co-optimization of S/D implantation and silicide process in the future.

## 4. Conclusions

In a summary, a four-levels vertically stacked Si_0.7_Ge_0.3_ NWs GAA device is successfully fabricated by developing a high-quality epitaxial growth and vertical fin etch of stacked SiGe/Si multilayer, and introducing the channel release process of stacked Si_0.7_Ge_0.3_ NWs under the optimal ACT@SG-201 solution. Meanwhile, the Ni-based silicide process is also implemented to improve its driven current by decreasing the R_SD_. Moreover, its slightly poor leakage performance needs to be further improved through the co-optimization of S/D implantation and silicide process in the future.

## Figures and Tables

**Figure 1 nanomaterials-12-00889-f001:**
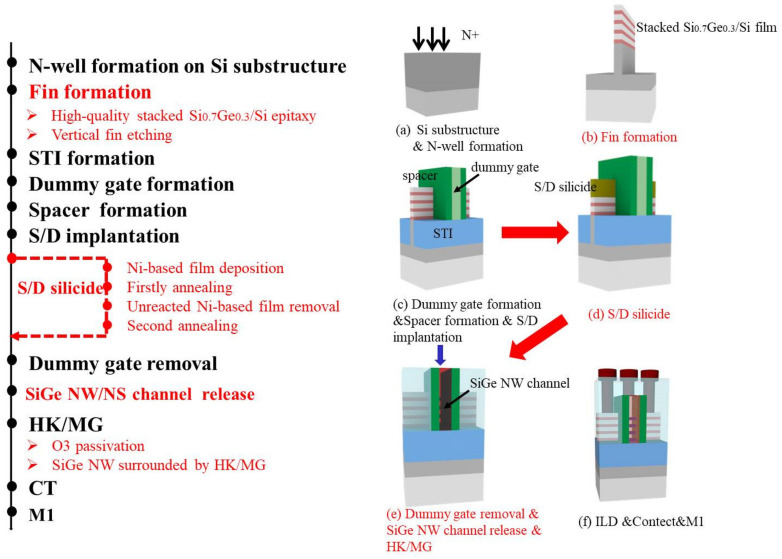
The process flow of stacked Si_0.7_Ge_0.3_ NWs GAA p-MOS device. (**a**) Si substructure & N-well formation; (**b**) fin formation; (**c**) dummy gate formation & spacer formation & S/D implantaton; (**d**) S/D formation; (**e**) dummy gate removal & SiGe NW channel release & HK/MG; (**f**) ILD & Contect & M1.

**Figure 2 nanomaterials-12-00889-f002:**
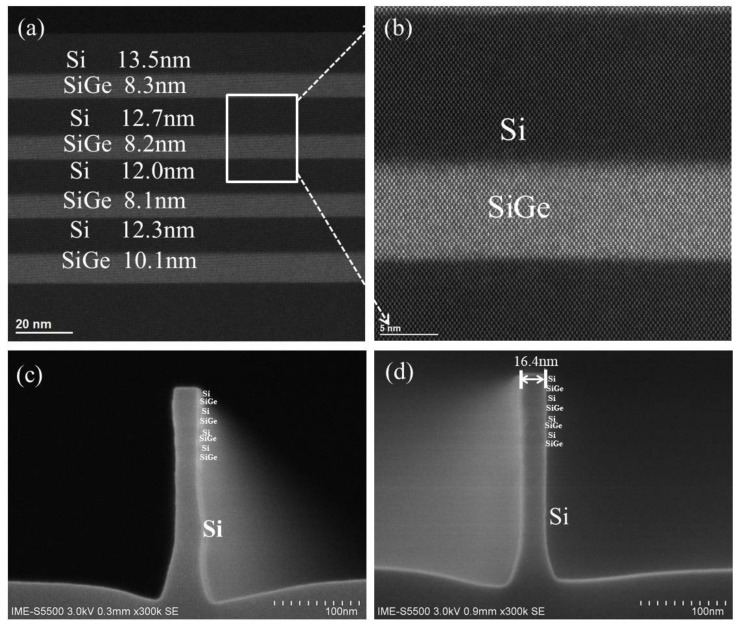
(**a**) HRTEM images of stacked Si_0.7_Ge_0.3_/Si multilayer; (**b**) its magnified images at the Si_0.7_Ge_0.3_/Si interfaces. (**c**) Four-period stacked SiGe/Si fin profile with HBr/O_2_/He plasma under bias voltage of −90 V, (**d**) under bias voltage of −100 V.

**Figure 3 nanomaterials-12-00889-f003:**
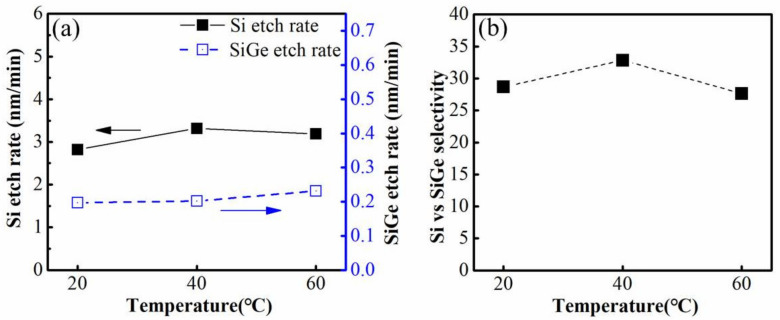
(**a**) The lateral etching rate of Si and the vertical etching rate of Si_0.7_Ge_0.3_, (**b**) the etching selectivity of Si to Si_0.7_Ge_0.3_ under ACT@SG-201 at different temperature.

**Figure 4 nanomaterials-12-00889-f004:**
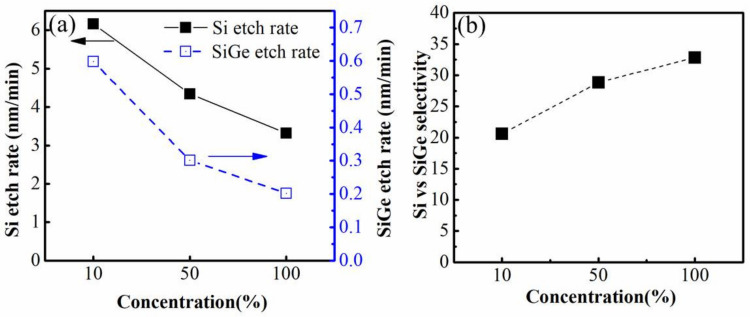
(**a**) The lateral etching rate of Si and the vertical etching rate of Si_0.7_Ge_0.3_, (**b**) the etching selectivity of Si to Si_0.7_Ge_0.3_ under different concentration of ACT@ SG-201 at 40 °C.

**Figure 5 nanomaterials-12-00889-f005:**
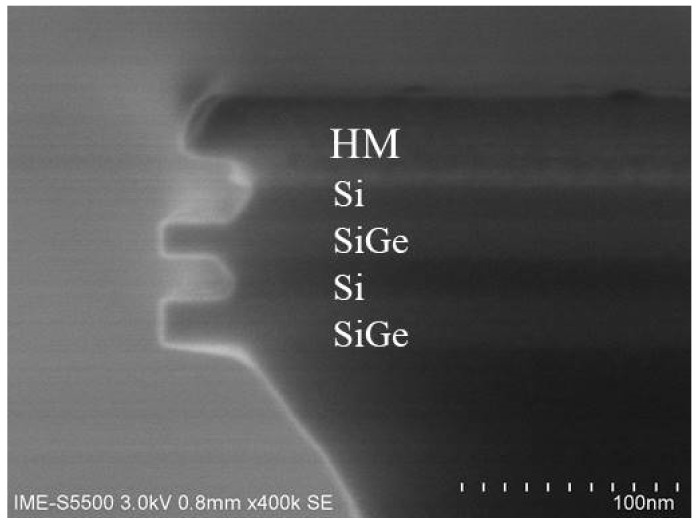
The SEM image of the selective etching of the Si_0.7_Ge_0.3_/Si multilayers stack utilizing the optimal ACT@ SG-201 solution at 40 °C for 10 min.

**Figure 6 nanomaterials-12-00889-f006:**
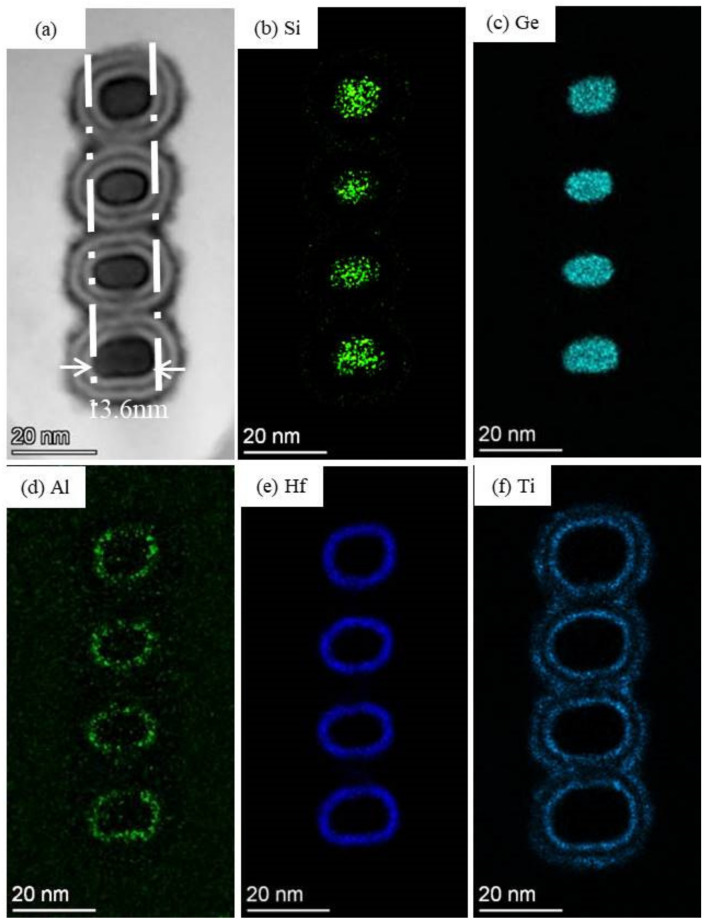
(**a**) The enlarged view of the channel area at across the fin direction and (**b**–**f**) the EDX mapping of the 4-levels vertically stacked Si_0.7_Ge_0.3_ NWs GAA device under the HK/MG stacks at the end of the fabrication processing.

**Figure 7 nanomaterials-12-00889-f007:**
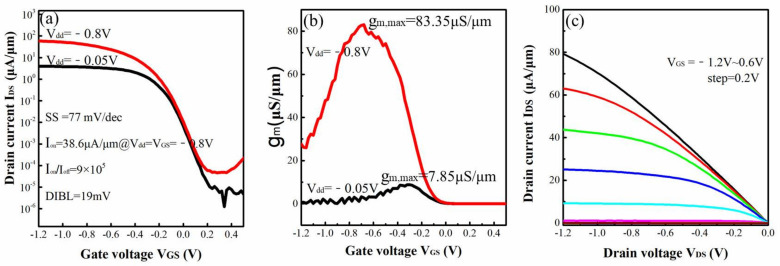
(**a**) I_DS_-V_GS_, (**b**) g_m_-V_GS_ and (**c**) I_DS_-V_DS_ characteristics of the four-levels vertically stacked Si_0.7_Ge_0.3_ NWs GAA device.

**Figure 8 nanomaterials-12-00889-f008:**
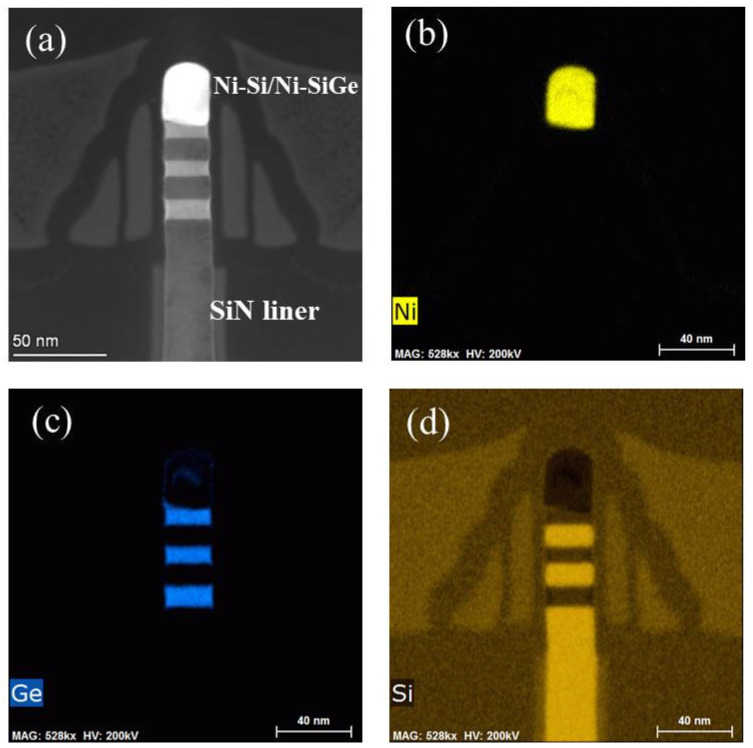
(**a**) The cross–sectional images of S/D region of 4-levels vertically stacked Si_0.7_Ge_0.3_ NWs GAA device at perpendicular to fin direction and its EDX mapping of Ni (**b**), Ge (**c**), and Si element (**d**) distribution.

**Figure 9 nanomaterials-12-00889-f009:**
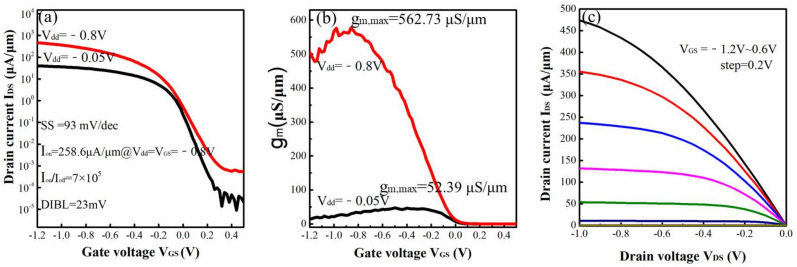
The (**a**) I_DS_-V_GS_, (**b**) g_m_-V_GS_ (**c**) I_DS_-V_DS_ characteristics of 4-levels vertically stacked Si_0.7_Ge_0.3_ NWs GAA device with S/D silicide process.

**Table 1 nanomaterials-12-00889-t001:** The comparison of electrical characteristic parameters for the 4-levels vertically stacked Si_0.7_Ge_0.3_ NWs GAA devices with or without S/D silicide process.

Samples	I_on_ (μA/μm)	SS (mV/dec)	g_m,max_ (μS/μm)	DIBL (mV/V)	I_off_ (μA/μm)	I_on_/I_off_	R_SD_ (Ω)
W/o silicide	38.6	77	83.85	19	4.22 × 10^−5^	9 × 10^5^	9.2 × 10^5^
W/silicide	258.6	93	562.73	23	3.71 × 10^−4^	7 × 10^5^	6.4 × 10^4^

## Data Availability

Not applicable.

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
