# Peer review of "4-Levels Vertically Stacked SiGe Channel Nanowires Gate-All-Around Transistor with Novel Channel Releasing and Source and Drain Silicide Process"

_nanomaterials, 2022, doi:10.3390/nano12050889_

Round 1

Reviewer 1 Report

This work is well organized, technically sound, clear and well written. However the topic and the argumentations are presented in a quite technical fascion and this affects detrimentally the potential broad interest of the work, making the manuscript under threshold for the Nanomaterials journal. I feel this work would instead be perfect for a slightly more technical journal published by MDPI.

Author Response

Dear Sir

We are very grateful to you for the comments of this work. Stacked nanowire or nanosheet is the most suitable new device architecture after FinFET for 3nm node and beyond since it offers excellent short channel control and effective channel width (Weff) per footprint. Moreover, to further improve the weak pFET performance, stacked nanowire or nanosheet with the high mobility of SiGe channel is a good candidate. So far, a cladded pFET SiGe NS device was explored by trimming the Si channel and growing Si1-xGex (x>0.3) after channel release (S. Mochizuki et al., IEDM., p2-3, 2020). Also, a compressively strained Si0.4Ge0.6 channel was fabricated on a Si0.7Ge0.3 strain-relaxed buffer, delivering high NS pFET hole mobility (A. Agrawal et al., IEDM., p2-2,2020.) However, no one has reported a SiGe channel formed during the initial epitaxial stack deposition. Therefore, we think this study is a valuable work and can be interesting for researchers dealing with the technology of high mobility channel based GAA transistors.

Best wishes

Yongliang Li

Reviewer 2 Report

Reference: nanomaterials-1574964

Review Report

The MS entitled “4-Levels Vertically Stacked SiGe Channel Nanowires Gate-All-Around Transistor with Novel Channel Releasing and Source and Drain Silicide Process” by Xiaohong Cheng et al. reports on the fabrication and characterization of 4-levels vertically stacked Si0.7Ge0.3 nanowire (NW) Gate All Around (GAA) p-MOS transistors. In the paper it is demonstrated the development of the fabrication process of high-quality stacked Si0.7Ge0.3/Si fins.

The English of the paper is rather good (nevertheless I advise a careful check by the authors to avoid typos) and the object of study is of genuine interest. In my opinion, authors must improve and extend the paper including more details about the electrical performance of the device.

Authors must answer the following questions:

1 – Are the SiGe layers under strain?

2 – I would suggest characterizing the quality of the layers using Raman.

3 – Figure 7 presents the output and transfer characteristics of a transistor: Were many samples measured or only one?

4 – Using the extracted source and drain resistances de-embed the output characteristics to obtain the intrinsic ones.

5 – Obtain and plot both the extrinsic and intrinsic transconductance characteristics.

6 – From point 5 extract and plot the efficiency of the transconductance.

7 – Compare results obtained 5 and 6 with the ones available in literature and/or with the expected ones from analytical models.

Author Response

Dear Sir

We are very grateful to the reviewer for the constructive comments, which are really helpful for improving the quality of our manuscript. We have taken into account all the comments made by the reviewers in this revision. The point-to-point responses to the comments by the reviewers are attached below for your considerations.

Best wishes!

Yongliang Li

1 – Are the SiGe layers under strain?

Reply: Based on the newly added HRXRD results of the stacked Si0.7Ge0.3/Si multilayer in Fig.2, SiGe layers are under strain. This is because that a series of obvious high intensity satellite peaks are found, indicating that the epitaxial layers of the stacked Si0.7Ge0.3/Si multilayer are under strain due to the lattice constant mismatch of Si and Si0.7Ge0.3. For details, please see the line 107th to 113th on page 3 in this revised version.

2 – I would suggest characterizing the quality of the layers using Raman.

Reply: Thanks for your suggestions. The Raman is a very good technology to characterize the quality of epi stacked layers, but our lab doesn’t have it so far. Therefore, we have added the HRXRD analysis results to check the quality of the layers. The high quality of the stacked Si0.7Ge0.3/Si multilayer is also conformed based on the HRXRD results. For details, please see the line 108th to 114th on page 3 in this revised version.

3 – Figure 7 presents the output and transfer characteristics of a transistor: Were many samples measured or only one?

Reply: Thanks for the reviewer’s comment. A total of 252 samples were measured on the same 8-inch wafer, and a representative transistor with a typical output and transfer characteristics is presented in this article. (in this revised version, Fig.7 has changed to Fig. 9).

4 – Using the extracted source and drain resistances de-embed the output characteristics to obtain the intrinsic ones.

Reply: Thanks for your suggestions. After extracting the source and drain resistances, the device will show the expected better performance. However, this action may lead to some deviations from the actual situation since the RSD can not be zero. In order to reduce the impact of high S/D resistance, we will do further study on the further reduction of source and drain resistances by employing the S/D epi process in the future.

5 – Obtain and plot both the extrinsic and intrinsic transconductance characteristics.

Reply: Thanks for your suggestions. We have updated the transconductance characteristics in Fig.9(b) and Fig11(b) for the device without and with the S/D silicide process. These two transconductance characteristics do not extract the source and drain resistances. For detail, pls see Fig.9 and Fig, 11 on page 8 and 10 in this revised version.

6 – From point 5 extract and plot the efficiency of the transconductance.

Reply: Thanks for the reviewer’s comment. We will do this study in the future because these are preliminary results and its electrical characteristics need to be further improved. Thanks for your consideration and understanding.

7 – Compare results obtained 5 and 6 with the ones available in literature and/or with the expected ones from analytical models.

Reply: Thanks for the reviewer’s constructive suggestion. So far, there are few literatures about the SiGe GAA device. So, there are few data to compare the transconductance of SiGe GAA devices. For example, we found that a SiGe GAA paper have do transconductance analysis, but only provide a relative transconductance value, not no absolute value. [for example, H. L. Chiang and T. C. Chen, et.al, 2020 VLSI] Moreover, this article presents the preliminary results of Stacked SiGe Channel Nanowires and its electrical characteristics need to be further improved in the future. We will do this comparison after the optimization of electrical performance. Thanks for your consideration.

In addition, we have modified some typos and have smoothed the English writing of the whole article. For details, please see the revised version. Thanks again for reviewer’s suggestion.

Reviewer 3 Report

no remarks

Author Response

Thanks for the reviewer’s comment.

Reviewer 4 Report

Good and interesting work. My only question is about the term "nanowires" : perhaps is more adequate to substitute it by "nanolayers".

Anyway, the paper is very well done. Good physics and engineering content. 

Author Response

Dear Sir

We are very grateful to you for the comments of this work. We still propose to call it as nanowires. Thanks again for your suggestion.

Best wishes

Yongliang Li

Round 2

Reviewer 1 Report

Well organized and technically sound, the manuscript unfortunately does not fulfill the impact and broad potential interest requested for publishing in Nanomaterials journal. Authors are encouraged to transfer the manuscript to more technical journals by MDPI.

Author Response

Thanks again for your comment.